# Functional and Clinical Characteristics for Predicting Sarcopenia in Institutionalised Older Adults: Identifying Tools for Clinical Screening

**DOI:** 10.3390/ijerph17124483

**Published:** 2020-06-22

**Authors:** Maria A. Cebrià i Iranzo, Anna Arnal-Gómez, Maria A. Tortosa-Chuliá, Mercè Balasch-Bernat, Silvia Forcano, Trinidad Sentandreu-Mañó, Jose M. Tomas, Natalia Cezón-Serrano

**Affiliations:** 1Department of Physiotherapy, University of Valencia, 46010 Valencia, Spain; angeles.cebria@uv.es (M.A.C.iI.); merce.balasch@uv.es (M.B.-B.); trinidad.sentandreu@uv.es (T.S.-M.); natalia.cezon@uv.es (N.C.-S.); 2Hospital Universitari i Politècnic La Fe, 46026 Valencia, Spain; sforcanosanjuan@gmail.com; 3Physiotherapy in Motion, MultiSpeciality Research Group (PTinMOTION), University of Valencia, 46010 Valencia, Spain; 4Research Unit in Clinical Biomechanic (UBIC), University of Valencia, 46010 Valencia, Spain; 5Department of Applied Economics, University of Valencia, 46022 Valencia, Spain; angeles.tortosa@uv.es; 6Psychological Development, Health and Society (PSDEHESO), University of Valencia, 46022 Valencia, Spain; 7Advanced Research Methods Applied to Quality of Life promotion (ARMAQoL), University of Valencia, 46010 Valencia, Spain; Jose.M.Tomas@uv.es; 8Department of Methodology for the Behavioural Sciences, University of Valencia, 46010 Valencia, Spain

**Keywords:** sarcopenia, older adults, institutionalised, functionality, clinical

## Abstract

Background: Recently, the European Working Group on Sarcopenia in Older People (EWGSOP2) has updated the sarcopenia definition based on objective evaluation of muscle strength, mass and physical performance. The aim of this study was to analyse the relationship between sarcopenia and clinical aspects such as functionality, comorbidity, polypharmacy, hospitalisations and falls in order to support sarcopenia screening in institutionalised older adults, as well as to estimate the prevalence of sarcopenia in this population using the EWGSOP2 new algorithm. Methods: A multicentre cross-sectional study was conducted on institutionalised older adults (n = 132, 77.7% female, mean age 82 years). Application of the EWGSOP2 algorithm consisted of the SARC-F questionnaire, handgrip strength (HG), appendicular skeletal muscle mass index (ASMI) and Short Physical Performance Battery (SPPB). Clinical study variables were: Barthel Index (BI), Abbreviated Charlson’s Comorbidity Index (ACCI), number of medications, hospital stays and falls. Results: Age, BI and ACCI were shown to be predictors of the EWGSOP2 sarcopenia definition (Nagelkerke’s R-square = 0.34), highlighting the ACCI. Sarcopenia was more prevalent in older adults aged over 85 (*p* = 0.005), but no differences were found according to gender (*p* = 0.512). Conclusion: BI and the ACCI can be considered predictors that guide healthcare professionals in early sarcopenia identification and therapeutic approach.

## 1. Introduction

Population aging is a worldwide phenomenon which has been expressive and accelerated over the years [1]. Therefore, geriatric syndromes, such as sarcopenia with a higher prevalence among institutionalised older adults (14–33%) than those living in community (1–29%) [2,3], has a considerable clinical and research interest [4].

The most widely used definition of sarcopenia is the one of the European Working Group on Sarcopenia in Older People (EWGSOP) [5,6], which was updated in 2018 (EWGSOP2) and focuses on low muscle strength as a key characteristic of sarcopenia, uses detection of low muscle quantity and quality to confirm the diagnosis and identifies poor physical performance as indicative of severity [7]. Sarcopenia has also been associated with increased risk of falls, impaired ability to perform activities of daily living and, consequently, it can cause functional dependency and disability in older adults [5,8]. However, most of this previous research has mainly focused on older adults living in the community rather than on institutionalised people [3]. Taking into account that strong evidence predictors of institutionalisation in older adults are functional impairment, cognitive impairment, higher age, low self-related health status and a high number of prescriptions [9], it could be suggested that older adults with sarcopenia living in institutions could even have a higher prevalence of some of these adverse outcomes related to sarcopenia, such as functional capacity impairment, dependence, falls, physical disability, negative impact on quality of life, hospitalisation and even death [10,11,12,13].

Assessment of sarcopenia needs measurement instruments for objective evaluation of muscle strength, mass and function, which also requires substantial time. Hence, screening of sarcopenia with user friendly, simple tools is required [14]. Clinical aspects and measurements such as functionality, comorbidities, number of drugs, number of hospitalisations and number of falls are widely used tools in residential facilities for assessing health and do not require specific and expensive medical equipment. However, the few studies that have focused on institutionalised people have shown isolated results of one or other characteristic, and not on a comprehensive perspective which is what defines a person’s health. Thus, functional capacity has been assessed mainly through the Barthel Index [15,16,17] and it has shown that participants diagnosed with sarcopenia tend to have worse functional status [15]. Others studies have identified the comorbidities of institutionalised people, quantifying the number of diseases [15,16,17,18] or using the Charlson’s Comorbidity Index [19]. Generally, those that relate it to sarcopenia show no differences in prevalence of diseases between the sarcopenic or non-sarcopenic groups [15,19]. Some studies have used variables such as number of drugs or number of hospitalisations in order to describe the participants [15,18] but no relationship has been established with sarcopenia. A number of falls have been quantified in institutionalised older adults with sarcopenia with no significant relationship [15]. For the moment, the variables that have been mainly associated with sarcopenia in institutionalised older people are the anthropometric ones (age, gender and body mass index) [15].

Therefore, some bivariate relationships have been studied between variables widely used in the clinical context and with sarcopenic institutionalised older people, but research in this area is still sparse. Moreover, the different combinations of functional and clinical variables in relation to sarcopenia in a multivariate framework remain to be elucidated. Thus, to the best of our knowledge, there is no study evaluating sarcopenia according to EWGSOP2 criteria and relating it with functionality, comorbidities, number of drugs, number of hospitalisations and number of falls in institutionalised older people in order to help clinicians in the screening and detection of sarcopenia in this population.

It was hypothesised that institutionalised older people suffering from sarcopenia as defined by the EWGSOP2 criteria and cut-off points would have a lower functional capacity, higher number of hospitalisations, higher number of drugs used, higher number of falls and higher index of comorbidities when analysed individually. Moreover, the combination of these factors may correlate higher to sarcopenia and help in the screening process.

The main aim of this study was to analyse the relationship between sarcopenia and functional ability, hospitalisation, number of falls, polypharmacy and comorbidity in order to support and facilitate sarcopenia screening in institutionalised older adults. A secondary aim was to identify which of our clinical and functional variables are the most relevant as supporting tools for screening sarcopenia and to estimate the prevalence of sarcopenia in institutionalised older adults using the new algorithm of the EWGSOP2.

## 2. Materials and Methods

### 2.1. Study Design

A multicentre cross-sectional study was carried out between January and November 2019 in institutionalised older adults living in the province of Valencia (Spain).

This study was approved by the Ethics Committee for Human Research of University of Valencia (H1542733812827) and was conducted in accordance with the Declaration of Helsinki. This research was registered in the ClinicalTrials.gov (ID: NCT03832608). All participants were briefed beforehand and all signed a written consent form before participating in the study.

### 2.2. Participants

The sample included adults institutionalised in residential facilities, aged 65 or older. The exclusion criteria were: (1) patients with edema that could alter the bioimpedance analysis (BIA) results; (2) Mini-Mental State Examination (MMSE) < 18 points [20]; (3) acute disease, hospital admission or unstable chronic disease in the last month.

### 2.3. Sarcopenia Definition

The EWGSOP2 has proposed an algorithm for case-finding, diagnosis and severity determination [7] which includes the SARC-F questionnaire, the measurement of muscle strength, muscle quantity or quality, and the identification of physical performance.

Following this algorithm, in this study the measured parameters to identify sarcopenia cases and its level of severity were:

The *SARC-F* is a 5-item questionnaire (strength, assistance walking, rise from a chair, climb stairs, and falls) based on cardinal features or consequences of sarcopenia, that allows to identify cases when the score is ≥ 4 points out of 12 for the total score [21].

*Muscle strength* (kg), was measured by the handgrip strength technique using a Jamar Plus+ digital hand dynamometer (Patterson Medical, Sammons Preston, Bolingbrook, IL, USA) [22]. Grip strength cut-off points for low strength were <27 kg and <16 kg for men and women, respectively [23].

*Muscle quantity* or Appendicular Skeletal Muscle Mass (ASM) was measured with BIA using the Bodystat^®^ 1500MDD (Bodystat Ltd., Douglas, UK). The device was calibrated daily using the standard control circuit supplied by the manufacturer. Before doing the test, participants were asked to follow these instructions [15]: (1) no previous physical exercise; (2) 2–3 h of fasting; (3) no alcohol or large amount of water intake; (4) urinating 30 min before; (5) every metal piece (such as a watch, jewellery) was taken off. Moreover, the test was not conducted in participants wearing a pacemaker and/or with edema [24]. The edema was assessed and diagnosed by the physician and recorded in clinical history. BIA test was done with the patient in supine position, on a non-conductive surface, ensuring that no parts of the body were touching. The patient stayed in this position for 5 min prior to measurement to ensure that fluid levels had stabilised in the body. Before placing the electrodes, the skin was cleaned with 70% alcohol. Using an ipsilateral tetrapolar method, the electrodes were placed behind the knuckle of the middle finger and on the wrist next to the ulna head (upper limb) and at the dorsal side of the second metatarsal head bone and on the ankle at the level of, and between, the medial and lateral malleoli (lower limb). The BIA was performed using an alternating sinusoidal electric current of 200 µA at a single operating frequency of 50 kHz. For estimating the ASM, the Sergi’s BIA equation was used: ASM (kg) = −3.964 + (0.227 × RI) + (0.095 × weight) + (1.384 × sex) + (0.064 × Xc) [25], where resistive index (RI) was resistance (ohms) normalised for height (cm), weight (kg), sex was 1 in men and 0 in women, and reactance (Xc, ohms). ASM cut-off points for low mass were <20 kg and <15 kg for men and women, respectively. The ASM Index (ASMI) was defined as ASM/height squared. ASMI cut-off points for corrected low mass were <7.0 kg/m^2^ and <5.5 kg/m^2^ for men and women, respectively [7].

*Physical performance* was assessed through gait speed (m/s) using a 4-m walking test [26], where <0.8 m/s was the cut-off point [5,27]. Participants had to walk at their usual walking speed and using their usual walking aid. Moreover, the physical performance was measured by the Short Physical Performance Battery (SPPB) [28], where ≤8 points was used as a cut-off point both in men and women.

Finally, sarcopenia was classified in different severity levels according the EWGSOP2 [7]: (1) probable sarcopenia when SARC-F scored ≥4 points and there was low muscle strength (grip strength <27 kg and <16 kg in men and women, respectively); (2) confirmed sarcopenia when also low quantity muscle was detected (ASMI <7.0 kg/m^2^ and <6 kg/m^2^ in men and women, respectively); and (3) severe sarcopenia, when confirmed sarcopenia was summed up to low physical performance (SPPB ≤ 8 point score).

### 2.4. Measurements

Added to the algorithm parameters established by the EWGSOP2, each participant underwent all of the following assessments on the same day. Different health professionals took these measurements. In order to avoid to inter-individual errors, intraclass correlation coefficients (ICCs) were calculated to know the interrater reliabilities. According to Koo and Li (2016) [29], values of ICCs between 0.75 and 0.9 indicate good reliability and values greater than 0.90 indicate excellent reliability. All ICCS in this study ranged from 0.802 to 0.985 which may be considered a very good reliability.

The studied variables were:

*Anthropometric variables*: (1) Age and gender; (2) body weight (kg), assessed using a Tanita BC 601 model weighing device (TANITA Ltd., Tokyo, Japan); (3) barefoot standing height (cm), measured with a stadiometer SECA 213 (Seca Ltd., Hamburg, Germany); body mass index (BMI), calculated based on the parameters of weight (kg) divided by height squared (m^2^).

*Functional ability evaluation* used the Barthel Index score [30]. This index was validated in older populations [31]. Values <20 points indicate total dependency for activities of daily living and scores between 21 and 60 indicate severe dependence [32].

*Comorbidity severity* was recorded using the Abbreviated Charlson’s Comorbidity Index [33]. It encompasses eight medical conditions (cerebral vascular disease, diabetes, chronic obstructive pulmonary disease, heart failure/ischemic heart disease, dementia, peripheral arterial disease, chronic kidney failure (dialysis) and cancer) with total scores ranging from 0–10, where 0 is no comorbidity and 10 is high comorbidity. On the other hand, this variable allows us to classify participants in relation to their comorbidity level (as the modified Abbreviated Charlson’s Comorbidity Index): absence of comorbidity is considered between 0 and 1 points, low comorbidity when the index is 2, and high comorbidity when it is ≥3 points.

*Number of medications* taken daily on a regular basis.

*Number of hospital stays* in the last year (recorded as the number of hospitalisations, either due to falls or any other clinical situation that required it).

*Falls* were analysed as number of falls in the last year (including both falls that did not require hospitalisation and those that required hospitalisation/surgery), and also registered according to the SARC-F questionnaire falls item (named as “modified falls”): no falls, 1–2 falls and ≥3 falls [21].

### 2.5. Sample Size Calculation

Given that the population size was larger than 100,000 and required accounting for the most variable situation (p = q = 0.5) with a confidence level of 95%, 375 subjects were needed. Of those, a stratification among institutionalised and community-dwelling adults were considered. For the purposes of this research, only institutionalised participants were considered.

### 2.6. Statistical Analyses

For descriptive purposes, the mean and standard deviation for quantitative variables were calculated, whereas percentages were estimated for categorical variables. At the bivariate level of the relationship, several inferential tests were performed. Specifically, when means from a quantitative variable wanted to be compared across the levels of a factor, ANOVAs or t-tests were employed. Assumptions for the correct use of these parametric techniques were previously tested and opportune corrections were applied if necessary. Partial eta-squares were obtained as measures of effect size. When two categorical variables were related, independence chi-square tests were used with Cramer’s V and Kendall’s tau as measures of the effect size. Finally, a binary logistic regression was used to predict the likelihood of having sarcopenia at the multivariate level. Beta coefficients as well as the odds-ratio associated with each predictor were calculated. Additionally, two estimates of the overall predictive power of the logistic regression were calculated: Cox and Snell and Nagelkerke’s R-square. Given the available sample size a stepwise procedure was used to select the predictors in the logistic regression. All statistical tests employed were considered statistically significant at *p* < 0.05 and in all cases, appropriate measures of effect size were estimated. All statistical analyses were performed in SPSS 24.

## 3. Results

### 3.1. Sample Characteristics

A total of 132 participants were included in this study. The age range for all the participants was 65–97 years, and according to gender the range was 65–96 years old and 65–97 years old for men and women, respectively. The mean age was 82 ± 8.3 years old and 102 participants (77.7 %) were women (Table 1).

Regarding the different cut-off points established by the EWGSOP2 algorithm, this study’s sample showed that men had a SARC-F mean under four points, whereas women’s mean was slightly over four. Grip strength values were just under cut-off points for men, and just over for women. Regarding ASM means in kg, both men and women scored below cut-off points, however, the ASM Index (kg/height^2^) means were just over cut-off points for both genders. In addition, muscle strength and quantity variables showed significant differences between gender (*p* < 0.001). Physical performance variables scored under cut-off points both for men and women, thus gait speed was below 0.8 m/s and SPPB was below eight points. In fact, both variables had means well below the cut-off points (0.56 ± 0.27 m/s for gait speed and 5.27 ± 2.99 score for SPPB). Moreover, the results of the SPPB indicate that 61% of the sarcopenic participants presented with severe sarcopenia.

In relation to the study variables, significant differences were found between genders for the two comorbidity-related variables measured. Women had significant lower mean than men in the Abbreviated Charlson’s Comorbidity Index (*p* < 0.01), and also the results of the modified Abbreviated Charlson’s Comorbidity Index for women indicated a better health status than men (*p* = 0.027). Moreover, these findings highlighted that women have less comorbidity than men (“No comorbidity” 55.5% vs. 33.3%, respectively) in this sample.

When all the steps of the EWSGOP2 algorithm were applied (Figure 1), the 132 institutionalised older adults were classified as follows: with no sarcopenia (n = 86, 65%), with probable sarcopenia which was not confirmed (n = 18, 14%), with confirmed sarcopenia (n = 0, 0%), and with confirmed- severe sarcopenia (n = 28, 21%).

### 3.2. Differences Based on EWSGOP2 Algorithm’s Application Regarding Study’s Variables

Regarding the severity levels of sarcopenia according to the EWGSOP2, results showed significant differences between the Barthel Index and the Abbreviated Charlson’s Comorbidity Index. The statistical descriptions are provided in Table 2. Post-hoc comparisons in the Barthel Index found statistically significant differences between the “no sarcopenia” and “probable sarcopenia” participants (*p* = 0.003) and between the “no sarcopenia” and “severe sarcopenia” participants (*p* < 0.001). Moreover, post-hoc comparisons in the Abbreviated Charlson’s Comorbidity Index found statistically significant differences between the “no sarcopenia” and “severe sarcopenia” participants (*p* = 0.004).

Results of the relationship between the functional and clinical variables with the sarcopenia severity levels are shown in Table 3. Although the effect size is low for all variables, results point out that non-sarcopenia participants were found among the independent and slightly moderate-dependent older adults, while participants with some degree of sarcopenia mainly had moderate or severe dependence. On the other hand, the non-sarcopenic older adults had less comorbidity and number of falls.

### 3.3. Derivation of the Regression Equation

A logistic binary regression was estimated to predict sarcopenia vs. non-sarcopenia. The two groups labeled with sarcopenia (probable or severe) were merged into a single sarcopenia group due to the sample size needed for stable estimates. The predictors considered were age, BMI and gender as control variables, together with Barthel’s Index, Abbreviated Charlson’s Comorbility Index, medications, hospitalisation stays and falls. Having taken into account previously presented bivariate results, the falls measure included in the regression was the indicator with three categories since it showed a better predictive power than the quantitative index. Due to sample size reasons an automated forward selection of predictors was used. Results of this logistic regression are shown in Table 4.

Three predictors had statistically significant effects on the dependent variable. The coefficients and associated odds-ratio for age showed that as age increases participants are more likely to have sarcopenia. The negative coefficient associated with the Barthel’s Index shows that, as was more likely, dependent people also have sarcopenia. Finally, the Abbreviated Charlson’s Comorbidity Index coefficient and odds-ratio showed that people with sarcopenia tend to present more comorbidities. This last index is the best predictor of the likelihood of presenting sarcopenia. Overall, the estimated effect size for the regression was 0.25 (Cox and Snell’s R-square) and 0.34 (Nagelkerke’s R-square).

### 3.4. Prevalence of Sarcopenia by Gender and Age

The overall prevalence of sarcopenia according to gender, including probable sarcopenia and severe sarcopenia cases, is presented in Figure 2. The difference between genders did not reach the significance threshold (χ^2^(2) = 1.33, *p* = 0.512, Cramer’s V = 0.101, Kendall’s tau = −0.039).

Table 5 shows the prevalence of sarcopenia according to age range. The participants aged over 85 years had a higher prevalence of sarcopenia, both for probable and severe sarcopenia (χ^2^(4) = 15.06, *p* = 0.005, Cramer’s V = 0.239, Kendall’s tau = 0.276).

## 4. Discussion

The present study showed that the Barthel Index, the Abbreviated Comorbidity Index and falls as registered in the SARC-F were significantly related with sarcopenia in institutionalised older adults using the EWGSOP2 algorithm. In addition, the Barthel Index, the Abbreviated Comorbidity Index and age of participants was shown to be able to predict sarcopenia in this population. Moreover, with the new algorithm, sarcopenia has been shown to be more prevalent in aged people.

To the best of our knowledge, the EWGSOP2 algorithm in institutionalised older adults has not been broadly used since the most recent definition [34]. This algorithm detects probable sarcopenia when low muscle strength is detected, and in clinical practice this is enough to start intervention [7], therefore it is highly important to identify institutionalised older adults in this category. This initial screening is done by finding the cases through the SARC-F, and only those identified by this tool have muscle strength assessed. In this regard, since all the parameters of the algorithm were analysed for all the participants, the descriptive data showed contradictory differences in the first two steps of the algorithm in relation to gender. The mean of SARC-F for men showed non-sarcopenic values, while women’s mean showed sarcopenic ones; however, the grip strength results were opposite. If using only the SARC-F for screening, this could be at the expense of missing men who would have been at least in the category of probable sarcopenia since they show low muscle grip strength, but are classified as not sarcopenic according to the SARC-F. Although the SARC-F has shown excellent specificity and has been widely used in the field of sarcopenia research in community-dwelling population [21,35,36,37,38], it has shown to have a major problem in relation to its low sensitivity. Previous research applying the EWGSOP definition has reported sensitivity to be 4.2% and 9.9% [35], or 14.6% and 33.3% [36] in men and women, respectively. The low sensitivity of SARC-F means that there is a high risk of a missed diagnosis of individuals who have sarcopenia, so other tools like the clinical and functional variables shown in this study may complement the SARC-F information and may be used in future research.

Following the EWGSOP2 algorithm, once the cases of probable sarcopenia have been detected the confirmation must be done with muscle quantity analysis. It has been stated that low muscle mass potentially contributes to disability and frailty in older adults [39]. Therefore, the accurate measurement of muscle mass is a crucial step for classifying sarcopenic older adults, especially in residential facilities since residents have higher risks of adverse events. There is an ongoing debate about the preferred adjustment for muscle mass indices and whether the same method can be used for all populations [7]. This is in line with our results, where the mean ASM (kg) was below cut-off points for both men and women, and the mean ASM Index (kg/height^2^) was over cut-off points for both genders. The EWGSOP2 consensus presented cut-off points for both ASM Index (kg/height^2^) [40] and ASM (kg) [27] for use when calculating muscle mass. Among these parameters, in the present study the ASM Index (kg/height^2^) was chosen since it consists of an anthropometric equation which adjusts through body size and has been reported to have associations with clinical outcomes such as physical disability, frailty or cardiovascular diseases [41,42,43]. However, it has to be taken into account that the most appropriate method defining low lean mass with the highest predictive value for identification remains uncertain [44]. Therefore, there is a need to elucidate in future research which method and operational definition is ideal for identifying sarcopenic people, especially older adults since they are at high risk.

In relation to the last step of the algorithm, 21% of the participants had severe sarcopenia which means that just over 60% of the sarcopenic people were in this category. Participants with severe sarcopenia not only have limited strength but also their performance is affected, which overall results in physical limitations [45] that can lead to adverse negative health outcomes such as care dependence, falls, fractures, hospitalisation and death.

Thus, in this study participants diagnosed with the EWGSOP2 algorithm had significantly lower functionality and higher comorbidity, especially those with severe sarcopenia. This supports previous studies which have also found a relationship between the Barthel Index and sarcopenia in institutionalised older adults [15,16] when applying the EWGSOP definition. Comorbidity has previously also been shown to be associated with sarcopenia, but it has been studied in community-dwelling people and measured by the presence of major chronic illnesses [46]. However, in the present study, severity of disease or comorbidity, which is an important issue in institutionalised older adults, has been carefully controlled by using the Abbreviated Charlson’s Comorbidity Index [33].

Furthermore, falls measured through the SARC-F were shown to be significantly related with sarcopenia in our population. However, in previous studies which used the EWGSOP definition with institutionalised older adults with sarcopenia, no significant relationship was found [15]. Overall, this seems to indicate, as has been previously stated, that the EWGSOP2 algorithm appears to be more sensitive than the EWGSOP for predicting the incidence of falls, although this has been shown in community-dwelling people [47]. This is clinically important since falls in older adults are a major cause of injury that may result in fracture, disability, poor quality of life, and death [48].

The trend that can be observed among sarcopenic people of this study is that medication and hospitalisations have more presence with both probable and severe sarcopenic participants. Thus, it was observed that both men and women present polypharmacy [49,50]. However, there was no significant relation with sarcopenic participants nor with hospitalisations, and although it has been previously stated that the risk of hospitalisation is higher in sarcopenic subjects [51], this has been studied in relation to community-dwelling people. This highlights the need to register these variables by a common and objectifiable tool to avoid differences in the registry protocols of each of the residential facilities.

In the regression analysis, the combination of age, Barthel Index and Abbreviated Charlson’s Comorbidity Index showed to be significant for predicting sarcopenia. These functional and clinical variables can help health care professionals in residential facilities to pay special attention to older people who may be heading towards suffering sarcopenia. A baseline sarcopenia assessment systematically carried out in residential facilities for new residents could provide important prognostic information regarding the patient’s future functional trajectory [52]. In this line, some authors have stated that the traditional medical model should move from a disease-centered perspective to a functioning-centered view [53]. Identifying loss of functionality and sarcopenia in early stages is important to prevent the progress of sarcopenia and its consequences, as well as to start treatment. Thus, treatment for sarcopenia is very important in residential facilities because the functional decline leads to a loss of independence in older adults and is associated with a higher demand for services in residential centers [54]. Therefore, the prevention of sarcopenia has become one of the major goals of public health professionals and clinicians [53], and easily applied tools for identifying it are of great importance.

As for the prevalence of sarcopenia, the application of the EWGSOP2 algorithm has shown that nearly 35% of the sample had some level of sarcopenia, which is within the wide range observed in previous studies of prevalence implemented in residential facilities up to the moment (17.7–73.3%) [54]. However, most of the studies carried out in this population have followed the EWGSOP algorithm [3,15]. Due to the novelty of the EWGSOP2 consensus, only one previous study has applied it in residential facilities [34] and showed a prevalence of 60%. Considering that their inclusion criteria was people aged 70 or more (higher age than our criteria) and that consequently their participants had a mean age of 85 years which is slightly higher than in the present study, and that sarcopenia is related to age [15], this can partially explain it. Future research with the EWGSOP2 algorithm in institutionalised older people could help ascertain the trends.

The EWGSOP2 definition classified the sarcopenic patients depending on the physical performance. The different categories showed that people with sarcopenia were mostly the older ones, and they were mainly diagnosed with severe sarcopenia. This is in line with other studies [15,55] which highlight the fact that in residential facilities people tend to have more dependency and disabilities [56].

Studies that have used the EWGSOP definition have found differences between gender but with contradictory conclusions. Some studies have shown women having a higher prevalence of sarcopenia than men (81.4% of sarcopenic patients), and in other studies—like the one conducted by Landi et al.—a higher ratio of sarcopenia corresponded to men [24]. When using the EWGSOP2 algorithm, no differences were found.

Taking into account that, currently, most health care professionals lack guidance or training to recognise and manage the decline in physical capacities in older age [45], our study offers promising results in relation to the assessment of sarcopenia with simple and available tools, such as the Barthel Index and Abbreviated Charlson’s Comorbidity Index.

### Limitations and Strengths

The main limitation of our study, common to other studies in residential facilities, is related to the inclusion criteria, which can exclude people with a greater probability of having sarcopenia and underestimate its prevalence. Another limitation is that the majority of the sample were women, and although this is characteristic related to aged population in Spain, greater equality between the sexes and their ages would be important in future research. This study offers the novelty of applying all of the steps of the updated definition of sarcopenia, plus the definition was applied on an institutionalised population, which has not been so broadly studied. Moreover, our results offer health professionals a new use to well-known tools that can support the identification of sarcopenia.

## 5. Conclusions

A functional tool, such as the Barthel Index widely used in residential facilities, and a clinical and objective index, such as the Abbreviated Charlson’s Comorbidity Index, can be considered predictors that guide healthcare professionals. This may support early sarcopenia identification and therapeutic approaches. Future research, with greater sample sizes and equality between gender and ages, may elucidate which of the different options of the EWGSOP2 algorithm may be more sensible to detect sarcopenic people, both in institutionalised and community-dwelling older adults.

## Figures and Tables

**Figure 1 ijerph-17-04483-f001:**
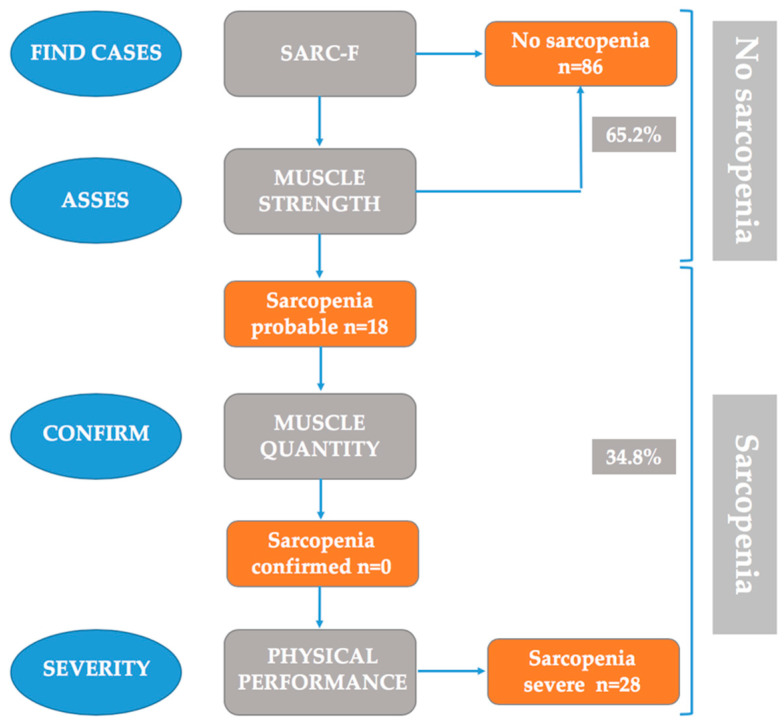
Sarcopenia: EWSGOP2 algorithm for case-finding, diagnosis and quantification of severity in practice.

**Figure 2 ijerph-17-04483-f002:**
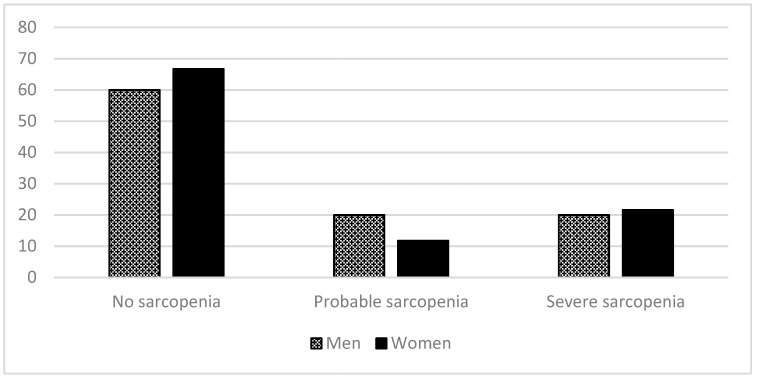
Percentages of no sarcopenia and severity sarcopenia levels (EWSGOP2, 2019) according to gender: no sarcopenia, probable sarcopenia, and severe sarcopenia.

**Table 1 ijerph-17-04483-t001:** Characteristics of the participants according to gender: mean ± standard deviation and (95% confidence interval) or number of cases (percentages).

Variables	Total (n = 132)	Men (n = 30)	Women (n = 102)	*p*-Value ^a^
**Anthropometrics**				
Age (years)	82.03 ± 8.25	78.70 ± 8.73	83.00 ± 7.88	0.11
	(80.61–83.45)	(75.44–81.96)	(81.46–84.56)	
Weight (kg)	66.66 ± 13.45	75.98 ± 12.60	63.92 ± 12.47	<0.001 ^†^
	(64.34–68.97)	(71.28–80.69)	(61.47–66.37)	
Height (cm)	154.02 ± 9.08	165.05 ± 8.00	150.77 ± 6.46	<0.001 ^†^
	(152.46–155.58)	(162.07–168.04)	(149.50–152.04)	
BMI (kg/m^2^)	28.06 ± 4.89	27.92 ± 3.83	28.10 ± 5.17	0.831
	(27.22–28.90)	(26.49–29.35)	(27.09–29.11)	
**EWSGOP2 algorithm**				
SARC-F (0–10 score)	3.95 ± 2.59	3.63 ± 2.77	4.04 ± 2.54	0.453
	(3.50–4.39)	(2.60–4.67)	(3.54–4.54)	
Grip strength (kg)	18.77 ± 7.82	26.85 ± 9.89	16.39 ± 5.10	<0.001 ^†^
	(17.42–20.11)	(23.16–30.55)	(15.39–17.39)	
ASM (kg)	15.10 ± 3.48	19.63 ± 3.14	13.84 ± 2.33	<0.001 ^†^
	(14.50–15.71)	(18.41–20.85)	(13.38–14.30)	
ASMI (kg/m^2^)	6.32 ± 0.98	7.20 ± 0.83	6.07 ± 0.87	<0.001 ^†^
	(6.15–6.49)	(6.87–7.52)	(5.90–6.24)	
Gait speed (m/s)	0.56 ± 0.27	0.57 ± 0.29	0.56 ± 0.27	0.797
	(0.51–0.61)	(0.46–0.68)	(0.50–0.61)	
SPPB (0–12 score)	5.27 ± 2.99	6.17 ± 2.84	5.00 ± 2.99	0.06
	(4.75–5.78)	(5.11–7.23)	(4.41–5.59)	
**Study’s variables**				
Barthel Index (0–100 score)	77.95 ± 19.07	79.00 ± 22.87	77.65 ± 17.92	0.767
	(74.67–81.24)	(70.46–87.54)	(74.13–81.17)	
Barthel Index classification				0.064
Independent (100)	23 (17.4%)	9 (30%)	14 (13.8%)	
Mild dependence (91–99)	11 (8.3%)	3 (10%)	8 (7.8%)	
Moderate dependence (61–90)	76 (57.6%)	11 (36.7%)	65 (63.7%)	
Severe dependence (21–60)	20 (15.2%)	7 (23.3%)	13 (12.7%)	
Total dependency (0–20)	2 (1.5%)	0 (0%)	2 (2.0%)	
Abbreviated Charlson’s	1.70 ± 1.33	2.27 ± 1.34	1.53 ± 1.29	<0.01 *
Comorbidity Index (0–10)	(1.47–1.93)	(1.77–2.77)	(1.28–1.78)	
Modified abbreviated Charlson’s Comorbidity Index ^b^				0.027 *
No comorbidity	66 (50%)	10 (33.3%)	56 (55.5%)	
Low comorbidity	35 (26.5%)	8 (26.7%)	27 (26.7%)	
High comorbidity	30 (22.7%)	12 (40.0%)	18 (17.8%)	
Medication (n)	8.67 ± 4.37	9.00 ± 4.59	8.58 ± 4.32	0.644
	(7.92–9.43)	(7.29–10.71)	(7.73–9.43)	
Hospitalisation stays (n)	0.24 ± 0.59	0.23 ± 0.50	0.25 ± 0.62	0.924
	(0.14–0.34)	(0.05–0.42)	(0.12–0.37)	
Falls (n)	1.13 ± 2.08	0.93 ± 1.48	1.19 ± 2.22	0.56
	(0.77–1.49)	(0.38–1.49)	(0.75–1.62)	
Modified falls (%) ^c^				0.773
No falls	65 (49.2%)	15 (50%)	50 (49%)	
1–2 falls	59 (44.7%)	14 (46.7%)	45 (44%)	
≥3 falls	8 (6.1%)	1 (3.3%)	7 (7%)	

Abbreviatures: BMI = Body Mass Index; SPPB = Short Physical Performance Battery; ASM = Appendicular Skeletal Muscle Mass; ASMI = ASM Index. ^a^
*p*-value unpaired Student’s t-test for quantitative variables and Chi-squared tests for qualitative variables; ^b^ Modified Charlson’s Comorbidity Index as a codification of total score in three comorbidity levels (Berkman et al., 1992) [33]; ^c^ Modified falls according to its registration through the SARC-F questionnaire by Malmstrom and colleagues (2016) [21]. * *p* < 0.05; ^†^
*p* < 0.01.

**Table 2 ijerph-17-04483-t002:** Means, standard deviation and ANOVA results of the independent variables.

Variables	EWSOP2 Algorithm	Mean ± SD	*F*	*df*	*df (error)*	*p*-Value	η^2^
Barthel Index (0–100 score)	NS	83.26 ± 16.90	10.992	2	129	<0.001 ^†^	0.146
	PS	67.78 ± 18.96					
	SS	68.21 ± 19.54					
Abbreviated Charlson’s Comorbidity Index (0–10)	NS	1.42 ± 1.30	6.054	2	129	0.003 ^†^	0.086
	PS	2.06 ± 1.26					
	SS	2.32 ± 1.25					
Medication (n)	NS	8.43 ± 4.57	0.561	2	129	0.572	0.009
	PS	9.61 ± 4.07					
	SS	8.82 ± 3.94					
Hospitalisation stays (n)	NS	0.17 ± 0.47	1.876	2	129	0.157	0.028
	PS	0.44 ± 1.04					
	SS	0.32 ± 0.55					
Falls (n)	NS	1.07 ± 2.39	0.44	2	129	0.65	0.007
	PS	1.56 ± 1.72					
	SS	1.04 ± 1.0					

Abbreviatures: SD = standard deviation; *F* = result of the F test; *df* = degrees of freedom; *η*^2^ Partial = partial eta-squared effect size; *p* = significance, ^†^
*p* < 0.01; NS = no sarcopenia; PS = probable sarcopenia; SS = severe sarcopenia.

**Table 3 ijerph-17-04483-t003:** Number of cases (n) and Chi-squared results of the independent variables.

Variables	EWSOP2			χ^2^	*df*	*p*-Value	Cramer’s V	Kendall’s τ
	Algorithm							
	NS	PS	SS					
Barthel Index classification				23.941	8	0.003 ^†^	0.301	−0.353
Independent (100)	22	1	0					
Mild dependence (91–99)	9	0	2					
Moderate dependence (61–90)	48	11	17					
Severe dependence (21–60)	6	6	8					
Total dependency (0–20)	1	0	1					
Mod-Abb-Charlson-Index ^a^				12.86	4	0.014 *	0.222	0.285
No comorbidity	52	6	8					
Low comorbidity	20	6	9					
High comorbidity	13	6	11					
Modified falls ^b^				14.87	4	0.005 ^†^	0.237	0.244
No falls	52	5	8					
1–2 falls	28	12	19					
≥3 falls	6	1	1					

Abbreviatures: NS = no sarcopenia; PS = probable sarcopenia; SS = severe sarcopenia; χ^2^ = result of Chi-square test; *df* = degrees of freedom; *p* = significance, * *p* < 0.05; ^†^
*p* < 0.01; V = Cramer’s V; T = Kendall’s Tau; ^a^ Mod-Abb-Charlson-Index = Modified Charlson’s Comorbidity Index as a codification of total score in three comorbidity levels (Berkman et al., 1992) [33]; ^b^ Modified falls according to its registration through the SARC-F questionnaire by Malmstrom and colleagues (2016) [21].

**Table 4 ijerph-17-04483-t004:** Binary Logistic Regression to predict sarcopenia vs. non-sarcopenia.

Variables	B	SE	*p*	Odd-Ratio	95% CI
Age	0.101	0.03	0.001	1.16	1.04–1.17
Barthel’s Index	−0.04	0.01	0.001	0.96	0.95–0.98
Abbreviated Charlson’s Comorbility Index	0.418	0.17	0.015	1.51	1.08–2.12

Abbreviatures: B = β coefficient; SE = standard deviation; *p* = significance; CI = Confidence interval.

**Table 5 ijerph-17-04483-t005:** Age-stratified sarcopenia prevalence: percentages and (n).

Sarcopenia Subtypes	65–74 Years	75–84 Years	≥85 Years	Total
No-sarcopenia (n = 86)	24.4% (21)	44.2% (38)	31.4% (27)	65.2%
Probable sarcopenia (n = 18)	16.7% (3)	16.7% (3)	66.7% (12)	13.6%
Confirmed sarcopenia (n = 0)	0%	0%	0%	0%
Severe sarcopenia (n = 28)	3.6% (1)	35.7% (10)	60.7% (17)	21.2%
n = 132	18.9% (25)	38.6% (51)	42.4% (56)	100%

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
