# Peer review of "Functional and Clinical Characteristics for Predicting Sarcopenia in Institutionalised Older Adults: Identifying Tools for Clinical Screening"

_ijerph, 2020, doi:10.3390/ijerph17124483_

Round 1

Reviewer 1 Report

This study was to analyse the relationship between sarcopenia and functional ability, hospitalisation, number of falls, polypharmacy and comorbidity in order to support and facilitate sarcopenia screening in older adults. And additional purpose is to identify which of our clinical and functional variables are the most relevant as supporting tools for screening sarcopenia.

It is well organized overall, but needs to be revised.

  1. It is thought that the number of study subjects is insufficient, and a description of the considerations for this is required.
  2. Detailed description of the BIA measurement method is required to support the results of muscle mass analysis by BIA measurement.
  3. Results of Table 1 and Table 5 were different for the age group of the study subjects. Given the ages listed in Table 1, the logistic results in Table 4 are not well understood.
  4. In the results of Sarcopenia evaluation, the results of physical performance are important. How was this considered?
  5. In the evaluation results of this study, there were very few subjects corresponding to sarcopenia, and the validity of the results obtained was questionable.
  6. When looking at the aspect of this study, it seems that the statistical processing method described in the research method is not sufficiently described.

Author Response

Response to Reviewer 1 Comments: Please see the attachment.

Reviewer 2 Report

The manuscript Functional and clinical characteristics for predicting Sarcopenia in institutionalised older adults: identifying tools for clinical screening by Cebrià i Iranzo et al is well presented and will add to the already existing literature on the topic of sarcopenia and how to identify this complex age related muscle loss.

The main concern with the study is the imbalance cohort. The majority of the subject involved are women (77.7%). This needs to be addressed as a limitation.

Additionally, these women had near 5 year age difference (older) to the men involved with the higher age range in men is actually the lower age range in women. Therefore the results may be obscured to a degree. if possible a subgroup of men and women with matched age group should be included to make sure that the gender related differences and age really do exist. If that cannot be done, then it is also needs to be addressed as a limitation. 

Author Response

Response to Reviewer 2 Comments: Please see the attachment

Round 2

Reviewer 1 Report

Overall, it has been revised well according to the judging results.